# *N*-Acetylation of Amines in Continuous-Flow with Acetonitrile—No Need for Hazardous and Toxic Carboxylic Acid Derivatives

**DOI:** 10.3390/molecules25081985

**Published:** 2020-04-23

**Authors:** György Orsy, Ferenc Fülöp, István M. Mándity

**Affiliations:** 1Institute of Pharmaceutical Chemistry, University of Szeged, Eötvös u. 6, H-6720 Szeged, Hungary; orsy.gyorgy@ttk.hu; 2MTA TTK Lendület Artificial Transporter Research Group, Institute of Materials and Environmental Chemistry, Research Center for Natural Sciences, Hungarian Academy of Sciences, Magyar Tudosok krt. 2, 1117 Budapest, Hungary; 3Research Group of Stereochemistry of the Hungarian Academy of Sciences, Dóm tér 8, H-6720 Szeged, Hungary; 4Department of Organic Chemistry, Faculty of Pharmacy, Semmelweis University, Hőgyes Endre u. 7, H-1092 Budapest, Hungary

**Keywords:** flow chemistry, acetylation, acetonitrile, safe, green chemistry

## Abstract

A continuous-flow acetylation reaction was developed, applying cheap and safe reagent, acetonitrile as acetylation agent and alumina as catalyst. The method developed utilizes milder reagent than those used conventionally. The reaction was tested on various aromatic and aliphatic amines with good conversion. The catalyst showed excellent reusability and a scale-up was also carried out. Furthermore, a drug substance (paracetamol) was also synthesized with good conversion and yield.

## 1. Introduction

*N*-acetylation is a widely used chemical reaction in general organic chemistry to build an acetyl functional group on an amine compound [1,2,3,4]. The use of the acetyl functional group is widespread, including drug research, the preparation of pharmaceuticals, polymer chemistry and agrochemical applications [5,6,7,8,9,10]. It can be utilized as a protecting group in many organic reactions and also in peptide synthesis [11]. In addition, it plays a major regulatory role in post-translational protein modification and regulation of DNA expression in all life forms [12,13].

In general, common acetylation reagents such as acetic anhydride and acetyl chloride, are easily accessible in chemical laboratories. Even the most sustainable technologies utilize these reagents in combination with various Lewis acids [14,15,16] and/or in neat form [17]. Nevertheless, the utilization of acetic anhydride and acetyl chloride have various drawbacks. Both reagents are major irritants and acetyl chloride is considered to be a genotoxic agent [18]. As such, the elimination of their use is of considerable current interest. 

Flow chemistry technology is widely used in many synthetic organic reactions at both laboratory and industrial scale [19,20,21,22,23,24,25,26,27,28,29,30,31,32,33,34,35]. There are a number of benefits of using continuous-flow (CF) chemistry. A wide range of reactions are much faster in flow processes, and fewer substrates and reagents are required [36,37,38,39,40]. Furthermore, more efficient and selective reaction can be carried out in continuous systems than in regular batch operations [41,42,43,44,45,46]. Additionally, flow reaction conditions can enable reaction routes that would otherwise only be feasible under regular batch conditions, e.g., higher temperature and pressure than can be used under safe conditions [47,48,49,50,51,52,53,54,55,56,57].

Whilst acetonitrile is a common solvent and is generally used in various fields of chemistry as eluent [41] and polar aprotic organic solvent [58], it is rarely used as reagent in organic chemistry. A few studies on acetonitrile as an acylation agent have been reported [59,60,61,62,63,64] thus far, for example Saikia et al., [65] presenting an unusual attempt to synthesize *N*-acylated aromatic amines. Acetonitrile was utilized as reagent and solvent with several Lewis acids (e.g., Cu(OAc)_2_, Mn(OAc)_2_, FeCl_3_, InCl_3_). The most promising catalyst was the trimethylsilyl iodide (TMSI), which activated the acetonitrile by co-ordination. On the other hand, Brahmayya et al. [66] developed a new method for preparing *N*-acetamides with metal-free sulfonated reduced graphene oxide catalyst under sonication.

Herein, we present the efficient utilization of acetonitrile for acetylation. We describe a selective and environmentally friendly CF acetylation of aromatic and aliphatic amines by the use of a low-cost and environmentally friendly Lewis acid catalyst as alumina in acetonitrile solvent with excellent conversions. These observations may open a new and green procedure for the synthesis of acetylated aromatic and aliphatic amines.

## 2. Results and Discussion

The reactions were carried out in a home-made CF reactor (See Appendix A). Our equipment consists of an HPLC pump that transports the substrate dissolved in acetonitrile. The solution is feed into a fillable HPLC column where the reaction occurs. The column is filled with solid catalyst. Additionally, there is a GC oven and an in-line back pressure regulator in the system to control the temperature and pressure of the reaction. A schematic outline of the reactor used in this study is shown in Figure 1.

In order to find the most useful solid Lewis acid catalyst for the flow synthesis, a reagent screen was carried out. Aniline, as a test compound was utilized, and acetonitrile was used both as solvent and acyl donor. The complete study of Lewis acids is shown in Table 1. The study shows that the most promising Lewis acid was the aluminum(III) oxide, other catalysts offered lower yields or no amide product formation was observed. Thus, the further reaction parameter optimization was carried out with aluminum(III) oxide as solid catalyst.

The starting material was dissolved in acetonitrile in a concentration of 100 mM, using aluminum(III) oxide powder as catalyst at a temperature of 25 °C and a pressure of 10–100 bar with 0.1 mL min^−1^ flow rate and 27 min residence time. The reaction did not show any pressure dependency and a moderate conversion of 27% was observed without any real pressure dependence. The effect of temperature on the conversion rate was tested at 50 bar pressure. Raising the temperature to 100 °C initiated the product formation with 53% conversion. By applying a higher temperature, the conversion of the reaction improved remarkably. It was found that the optimal temperature is 200 °C. Further increase of the temperature resulted in lower conversion values. In this case, the acetonitrile was in its supercritical state (Tc = 275 °C, Pc = 48 bar) due to the significant solvent expansion produced under supercritical conditions. This is a likely explanation of the decrease of the observed yield above 275 °C. The results obtained for the temperature dependence are summarized in (Figure 2a). The effect of pressure on reaction outcome was also tested (Figure 2b). The results indicate that a modest pressure (50 bar) is needed to increase the reaction conversion, while further pressure increase did not influence the reaction outcome significantly. The flow rate was tested too (Figure 2c). The results show that the optimum flow rate is 0.1 mL min^−1^. Any increase in the flow rate above this value resulted in decreased conversion. The effect of concentration on the reaction outcome was also tested (Figure 2d). The results show that increased concentration results in the decreased conversion.

Under the optimized conditions, the scope of the reaction was explored by a variety of aniline derivatives (Table 2). The anilines containing either electron-donating or electron-withdrawing groups were selected. It must be noted that all reactions were carried out in a single run and the products were analyzed by ^1^H and ^13^C NMR spectroscopy. Column chromatography purification of the product was only needed for compound **6**. For the others, only a simple evaporation of acetonitrile was required.

Importantly, the hydroxyl group possessing aniline **3** was acetylated with excellent yield and the drug substance paracetamol **4** with quantitative yield after a simple recrystallization. Lower yield was observed for 4-methoxyaniline (**5**), and the acetylated product was isolated after column chromatography with a mere 51% yield. For the halogen atom possessing anilines (**7**, **9** and **11**) excellent yields were achieved. For nitroanilines (**13**,**15** and **17**) no conversion was observed and only the starting material was isolated. This fact might be explained by the highly electron-withdrawing nature of the nitro substituent, which reduces the nucleophilicity of the amino group. Besides aromatic amines, aliphatic primary and secondary amines were also tested. The primary benzylic amine (**23**) was converted to the corresponding acetamide with excellent yield. When secondary amine piperidine (**19**) and morpholine (**21**) were tested, the acetylated derivatives (**20, 22**) were isolated with quantitative yields. The use of this reaction in stereoselective reaction was also tested with acetylation reaction carried out for the two enantiomers of 1-phenylethanamine (**25** and **27**). For both enantiomers the acetylated derivative was achieved with quantitative yield and the complete retention of the enatiomerical purity. This was investigated by optical rotation and was found to be identical to literature data.

The catalyst reusability was also tested. It was an important finding that the activity of the catalyst did not decrease significantly until 10 cycles and one cycle was carried out with 20 mg of benzylamine. This excellent result [67] opened the way to scale up the reaction, which was tested with the same reaction. The acetylation process could be scaled up to 2 g of benzylamine without significant decrease of the conversion. The product (*N*-benzylacetamide, **24**) was isolated with 94% yield after a simple recrystallization. The reaction was completed within 28 hours. This is considerably faster than what has been reported with already known technologies. Results are shown in Figure 3.

These results can be explained by the proposed reaction mechanism (Scheme 1) which relies on literature data [63,64,65,66]. A key step is the coordination of the lone electron pair of the nitrogen atom of the cyanide group, which yields a positive charge. Due to mesomeric structures, the positive charge might be localized on the carbon atom of the cyanide group. This positively charged carbon atom might be attacked by the lone pair electron of the amine yielding amidine, which is further hydrolyzed to provide the acetamide as shown in Scheme 1. The origin of the water is the residual water content of the solvent and the alumina. The addition of extra amount of water decreased the conversion of the reaction.

## 3. Materials and Methods

### 3.1. General

All solvents and reagents were of analytical grade and used directly without further purification. Fe_2_O_3,_ Boric acid, AlCl_3_, Al_2_O_3_ (for chromatography, activated, neutral, Brockmann I, 50–200 µm, 60 A) catalysts used in this study were purchased from Sigma-Aldrich (Budapest, Hungary), while Acetonitrile (100, 0%) was HPLC LC MS-grade solvents from VWR International (Debrecen, Hungary). 

### 3.2. General Aspects of the CF Acetylation

The CF acetylation reactions were carried out in a home-made flow reactor consisting of an HPLC pump (Jasco PU-987 Intelligent Prep. Pump), a stainless steel HPLC column as catalyst bed (internal dimensions 250mm L × 4.6 ID × ¼ in OD), a stainless steel preheating coil (internal diameter 1 mm and length 30 cm) and a commercially available backpressure regulator (Thalesnano back pressure module 300™, Budapest, Hungary, to a maximum of 300 bar). Parts of the system were connected with stainless steel tubing (internal diameter 1 mm). The HPLC column was charged with 4 g of the alumina catalyst. It was then placed into a GC oven unit (Carlo Erba HR 5300 up to maximum a 350 °C). For the CF reactions, 100 mM solution of the appropriate starting material was prepared in acetonitrile. The solution was homogenized by sonication for 5 min and then pumped through the CF reactor under the set conditions. After the completion of the reaction, the reaction mixture was collected and the rest solvent was evaporated by a vacuum rotary evaporator. 

### 3.3. Product Analysis

The products obtained were characterized by NMR spectroscopy. ^1^H-NMR and ^13^C-NMR spectra were recorded on a Bruker Avance DRX 400 spectrometer (Bruker AVANCE, Billerica, MA, USA), in DMSO-*d_6_* as solvent, at 400.1 MHz. Chemical shifts (δ) are expressed in ppm and are internally referenced (^1^H NMR: 2.50 ppm in DMSO-*d_6_*). Conversion was determined via the ^1^H-NMR spectra of the crude materials. A PerkinElmer 341 polarimeter (PerkinElmer, Boston, MA, USA) was used for the determination of optical rotations.

## 4. Conclusions

We have developed a sustainable CF process for the selective N-acetylation of various amines. The obtained chemical process is time and cost efficient, as it utilizes cheap reagent and catalyst and considerably faster with a residence time of only 27 min. Importantly, the well-known, cheap and non-toxic Lewis acid alumina was used as catalyst. Moreover, the acetylation reagent was acetonitrile, which is an industrial side-product with a modest price and it is considerably milder than those for the regular carboxylic acid derivatives used for acetylation, e.g., acetyl chloride, acetic anhydride. In general, under the optimized conditions good or excellent conversions were observed for the aromatic amines, except for the nitro substituted compounds, where no conversion was reached. Importantly, the painkiller drug substance paracetamol was gained with high yield. Mainly complete conversions were also observed for primary and secondary aliphatic amines. The use of this acetylation in stereoselective reactions was also tested and during the course of the reaction no racemization was observed for either enantiomer of enantiomerically pure 1-phenylethanamine.

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
