# Peer review of "N-Acetylation of Amines in Continuous-Flow with Acetonitrile—No Need for Hazardous and Toxic Carboxylic Acid Derivatives"

_molecules, 2020, doi:10.3390/molecules25081985_

Round 1

Reviewer 1 Report

In this manuscript, Fulop and Mandity describes an in flow process for the N-acetylation of amines using CH3CN activated by the presence of Al2O3 .
Although the synthetic transformation is interesting, after carefully consideration, the reviewer does not think that this manuscript is suitable for publication on molecules in its form, for the following reasons.

1) The introduction of the manuscript is not well organized. Many references have been reported for flow chemistry applications, but no references on the use of nitriles as acyl equivalents are given. N-acetylation with CH3CN promoted with Lewis acids (Cu(OAc)2, Mn(OAc)2, FeCl3, InCl3) are indicated in the text, but references are missing.

2) In the discussion section, figure 1 indicates the presence of preheated coil of 30 cm, but no information in the text nor in the supporting information is given, as well as no detail (length, diameter etc) of the HPLC column used is reported. These are serious issues which compromise the reproducibility of the work.

3) No study on the concentration variation has been reported, and there is not a defined protocol of operations to follow (lines 75-77 – something is wrong).

4) Parameters such as space time yield and productivity have not been reported for the continuous flow process, so it is very difficult to evaluate the efficiency of the process.

5) The proposed mechanism on scheme 1 relies to some literature data which have not been reported.

6) The scope of the reaction is quite limited, and needs to be extended to other anilines bearing different substituents. I suggest to employ also other ortho-substituted anilines in order to see if this could be a  limit of the reported methodology. In addition, only one example of secondary aliphatic amine is shown (table 2, entry 10). What happens using with primary amines?

7) English needs to be improved, there are a lot of wrong grammar constructions  and typos.

In my opinion, an extensive revision of the manuscript is required before considering for publication.

Author Response

Our answers for Reviewer 1 are as follows:

  1. The required references were added.
  2. The exact parameters for the reactor parameters were given.
  3. The concentration dependence was measured and results were added to the manuscript text. A defined and extended protocol is implemented too in the supporting information.
  4. The space time yield values were added to Table 2.
  5. The references were added.
  6. To extend the scope of the reaction as further aniline the 3-fluoroaniline, as a secondary amine the piperidine was tested. The results are excellent. Concerning primary amines the benzylamine and the 2 enantiomers of 1-phenylethanamine gave excellent results.
  7. The grammar was checked and the typos were corrected.

Reviewer 2 Report

The authors propose a continuous-flow acetylation of amines, using acetonitrile as acetylating agent and alumina as catalyst. The reactions are carried out in an home-made reactor equipped with a preheater coil and a backpressure regulator. The approach result quite interesting, although you need to consider some aspects carefully.

1) The best catalyst is Al2O3 for chromatography, activated, neutral, Brockmann I, 50-200 µm, 60 A. In order to obtain further mechanistic informations and to compare the catalytic activity of neutral alumina, I suggest to try also the acidic and basic alumina.

2) To the staves 78 and 79, the authors report that “The reaction did not show any pressure dependency, a moderate conversion of 27% was observed without the observation of any real pressure dependence”. In addition, in fig. 2 description (line 91) it is written “The effect of the pressure was measured at room temperature”. Where is the graph that describes the pressure effect? The authors could go into more detail on this matter.

3) The scope of the reaction has been explored, but not extensively: some other example of anilines with electron-withdrawing groups could be appreciated by readers.

4) The authors say “ all reactions were carried out in a single run and the products were analyzed by 1H and 13C NMR spectroscopy after simple evaporation of acetonitrile”. What about the compound 6? It is the conversion complete? It is necessary a purification?

5) In the scheme 1, a carbamate is reported after the hydrolysis of amidine: what is a mistake? In addition, from a mechanistic point of view the authors proposed an important role of the water. I mean, where the water molecule come from? It is important to explain this point.

6) In the introduction part, recent articles and review would be added: a) Journal of Flow Chemistry, Volume 8, Issue 3-4, 1 December 2018, Pages 109-116;  b) Chemical Society Reviews, 2016, 45 (18), pp. 4892-4928. c) Beilstein Journal of Organic Chemistry, Volume 13, 14 March 2017, Pages 520-542; d) Journal of Flow Chemistry, Volume 9, Issue 4, 1 December 2019, Pages 231-236.

7) The title of the supplementary material section is not correct with respect to the manuscript.

8) It is possible to add in the supplementary material the picture of the home-made reactor?

9) A brief description of the used backpressure regulator would be given in the Supplementary information.

9) In the title of the manuscript, please correct the word carbocylic.

With this major changes, the article could be valuable for the publication in Molecules.

Best regards

Author Response

Our answers for Reviewer 2 are as follows

  1. The acidic and basic alumina was not tested, since the neutral gave satisfactory results. Moreover, it is not available now in the lab and our budget is locked due to the COVID-19 situation.
  2. The pressure dependence of the reaction was tested too, and the results are shown in Figure 2.
  3. As further substrate with electron withdrawing substituent, the 3 fluoroaniline was examined and it gave excellent result.
  4. Compound 6 was purified by column chromatography. This fact is now indicated in the text too.
  5. The origin of the water is the residual water content of the solvent and the alumina.
  6. The suggested papers were cited.
  7. The title in the supplementary information was corrected.
  8. The images are added to the supplementary information.
  9. A brief description of the used backpressure regulator is added to the supplementary information.
  10. The title was corrected.

Reviewer 3 Report

In this manuscript Orsy and coworkers present a continuous flow process for the acetylation of anilines and one amine using acetonitrile, alumina, and an HPLC oven. The researchers compare alumina to a few Lewis acids and optimize the conditions for the reaction. They show that a small number of amines are effective for the reaction, including the retention of chirality and the lack of side acetylation of compoundss with multiple nucleophilic functional groups.

The acetylation reaction itself is extremely pedestrian and the researchers do not adequately describe what the unsolved problem and the contribution are here. They correctly argue that the genotoxicity of acetyl chloride make it unattractive as an acetylating reagent. But then I checked their reference 18 and found that acetic anhydride is not mentioned as genotoxic. If a process based on acetonitrile would be more attractive from a cost basis or for the lack of purification steps, then I would like to see that explained. Also, there is not adequate discussion of how else this reaction can be performed. Is the acetylation with acetonitrile a novel reaction? If not, then is the lack of a continuous flow process a barrier to commercialization? The high temperatures and pressures given here do not seem particularly mild despite what they have claimed.

The substrate scope is limited to 12 examples, including 2 that are enantiomers. Their analysis of the results is problematic in a couple of regards. First, they report exclusively assay yields and it is not clear that they have even used internal standards for that. They do not mention that any of their yields have been confirmed by isolation, which makes me pretty suspicious. Their presentation is sloppy in a number of places. Table 1 shows a reaction scheme that should probably have been used for Figure 2, since it shows Al2O3 and p,T, while in Table 1 the Lewis acid is varied and the temperature and pressure are not. In Scheme 1 a carbamate is drawn where the N-arylacetamide should have been drawn. In Figure S17 an NMR is obscured by some graphical error. 

For me to endorse publication of this paper, it will need a lot of work. First of all, the authors should obtain isolated yields from all of their acetylation reactions. At the very least they need assay yields with an internal standard. Next they need to explain the contribution that this work makes in terms of the chemical process. Lastly, I would like to see them improve the presentation of the data.

Minor error: Line 43 the word "filed" should be replaced with "field"

Author Response

Our answers for Reviewer 3 are as follows:

  1. The argues of the reviewer were analyzed and clarified. The basic benefit of our method is that there is no need for aggressive, harmful and toxic reagents. We tried to further emphasize this fact in our manuscript. The genotoxicity problem was clarified only for the acetyl chloride. We state that our reagents are milder, and not our method, however reagents spend only 27 min at elevated temperature.
  2. We reported yields in Table2 and now the text is corrected. Where appropriate, the purification and isolation methods are described too. The reaction schemes in Table 1, Figure 2 and Scheme 1 is corrected. There was no problem with Figure S17 in our file.
  3. The required changes were fulfilled.

Round 2

Reviewer 1 Report

I would like to inform you that I am satisfied with the changes made on this new version of the manuscript. The quality of the manuscript was improved, and this new version satisfy all my previous observations.

Some typos are still present:

line 117,  "amine" should be replaced with "amines"

line 124,   "teste" should be replaced with "tested"

regards

Author Response

The typos were corrected.

Reviewer 2 Report

My thoughts are with them during the current COVID-19 crisis and lockdown in all over the world- I hope that they, their families and colleagues are safe. I strongly emphasise to the editor that further experiments are clearly out of the question in the near future, and where I have suggested these this is with a future full length publication of this work in mind, rather than a necessary revision for the publication of this manuscript.
The authors have provided a new version of the work where they reply enough to the referees' comments.

Just a correction in references 55 and 56: change the first author "Angelis" with "De Angelis".

In this form, the work could be published in Molecules.

Best regards

Author Response

The typos were corrected.

Reviewer 3 Report

This version of the text is a considerable improvement over the first, and largely answers the questions that I have posed. I still think considerable English editing is required, and here are some of my suggested edits:

Abstract. "prominently cheap and safe" and "prominently mild" -> just drop the word "prominently

26. "pharmaceutics" -> "pharmaceuticals"

50. "metal free sulfonated" -> "metal-free sulfonated"

114. "This fact might be answered" -> "This fact might be explained"

117. "As secondary amine ... were tested" --> "When secondary amines ... were tested"

124. "teste" -> "tested"

155. "except for the nitro-substituted compounds, were no conversion was reached" -> "except for the nitro-substituted compounds, where no conversion was observed"

155. "pain killer" -> "painkiller"

Author Response

The typos were corrected.